# Controlling pairing of π-conjugated electrons in 2D covalent organic radical frameworks via in-plane strain

Isaac Alcón [1✉], Raúl Santiago[2], Jordi Ribas-Arino[2], Mercè Deumal [2], Ibério de P. R. Moreira[2] & Stefan T. Bromley [2,3✉]

Controlling the electronic states of molecules is a fundamental challenge for future sub-nanoscale device technologies. π-conjugated bi-radicals are very attractive systems in this respect as they possess two energetically close, but optically and magnetically distinct, electronic states: the open-shell antiferromagnetic/paramagnetic and the closed-shell qui-noidal diamagnetic states. While it has been shown that it is possible to statically induce one electronic ground state or the other by chemical design, the external dynamical control of these states in a rapid and reproducible manner still awaits experimental realization. Here, via quantum chemical calculations, we demonstrate that in-plane uniaxial strain of 2D covalently linked arrays of radical units leads to smooth and reversible conformational changes at the molecular scale that, in turn, induce robust transitions between the two kinds of electronic distributions. Our results pave a general route towards the external control, and thus tech-nological exploitation, of molecular-scale electronic states in organic 2D materials.

[1] Institut für Chemie und Biochemie, Physikalische und Theoretische Chemie, Freie Universität Berlin, Berlin, Germany. [2] Departament de Ciència de Materials i Química Física & Institut de Química Teòrica i Computacional (IQTCUB), Universitat de Barcelona, Barcelona, Spain. [3] Institució Catalana de Recerca i Estudis Avançats (ICREA), Barcelona, Spain. ✉email: ialcon8@gmail.com; s.bromley@ub.edu

Since the discovery of graphene[1], an increasing number of inorganic 2D materials with a range of physical and chemical properties have been synthesised by top-down approaches (e.g. by exfoliation of layered bulk materials)[2]. Due to their unique combination of extreme thinness and extended planarity, these monolayered systems should represent ideal platforms for tailoring electronic properties through in-plane strain[3]. However, in practice, 2D inorganic materials possess very high in-plane tensile strengths, which only allow for moderate strains (<5%)[3] generating modest electronic changes[4–6]. In parallel, chemists have developed alternative bottom-up synthesis approaches based on linking arrays of molecular building blocks, to produce 2D covalent organic frameworks (2D-COFs)[7–9]. These materials are significantly more flexible than their inorganic analogues[10] and allow the integration of the vast structural and electronic versatility of organic molecular chemistry[9,11] within scalable and robust 2D platforms. Building on previous proposals to take advantage of these traits for applications[8,12,13], we show that a particular family of 2D-COFs may provide an ideal basis for highly sensitive strain-control of electronic properties.

Kekulé bi-radicals are promising systems for organic molecular electronics[14–17] and magnetism[18] due to their intrinsically bi-stable electronic structure whereby two very distinct but energetically close resonant forms compete: namely, the open-shell antiferromagnetic (AFM) and the closed-shell quinoidal diamagnetic states (Fig. 1a). The different character of these two states leads to significant observable differences in the respective associated magnetic[19], optical[20,21] and structural[22] properties. It has been shown that molecular chemical design is an effective way to induce one electronic state or the other[16], where the length of the π-conjugated bridge between radical centres (e.g. a number of benzene rings[19]) and its conformation (e.g. dihedral angles[21]) are key parameters. Taking advantage of the effect of temperature on structural conformation (e.g. bond vibrations, aryl ring rotations), molecules have been synthesised in which thermal changes can induce a transition between the two kinds of electronic states[23,24]. Compared to the response speeds of typical electronic components, heat flow is very slow and so the thermal operation of nanoelectronic devices is difficult to envision. Moreover, such developments are currently focused on discrete molecules, thus hindering the scale-up of this bi-stable switching capacity to the device scale.

Bypassing the difficulties in manipulating individual molecules, our proposal starts with the integration of bi-radicals into 2D covalent organic radical frameworks (2D-CORFs), as schematically shown in Fig. 1. This covalent integration permits the direct manipulation of the structural conformations of bi-radicals through the application of external forces to the material while preserving their well-defined and stable positions within the 2D framework[25]. Moreover, 2D-CORFs can exhibit multi-radical (e.g. FM[26–28] or AFM[29]) and diamagnetic quinoidal closed-shell

solutions[30], confirming that the electronic bi-stability from the bi-radical monomers persists in the resulting 2D materials. Recently, different examples of experimentally synthesised 2D-CORFs have been reported, exhibiting multi-radical electronic ground states with either AFM coupling[31,32] or weak coupling (i.e. paramagnetism) between spins[33].

Dihedral angles of aryl rings are effective conformational parameters to control the localisation/delocalisation of unpaired electrons in π-conjugated organic radicals[34] (e.g. triarylmethyls, TAMs) and so determine the balance between the AFM and quinoidal states in bi-radicals[18,21]. As we have recently shown, compression is an effective manner to homogeneously flatten all aryl rings in 2D-CORFs to significantly increase electron delocalisation[35]. However, to induce local electron pairing one needs to asymmetrically flatten only some aryl rings with respect to others. Previously, we predicted that such a type of conformational manipulation in paramagnetic 2D-CORFs is possible via in-plane uniaxial strain (arrows in Fig. 1b)[25,36]. Numerous ways to experimentally induce significant strain in a 2D material placed on a substrate have been proposed such as (i) growing the 2D material on epitaxial substrates with a controlled lattice constant mismatch, (ii) thermal-expansion mismatch between the 2D material and its substrate and (iii) transferring the 2D material onto a flexible substrate and directly stretching, compressing or bending the substrate[3,37]. Herein, we show how modest in-plane stretching of 2D-CORFs could be a very promising route to externally and sensitively control the balance between both electronic configurations, and thus modulate their resulting optical, magnetic and electrical response.

In this work, we study the effect of uniaxial in-plane strain on a series of 2D-CORFs, including the recently synthesised $PTM_{acetylenic}$ material[32], by means of density functional theory (DFT) based calculations. Our results demonstrate that uniaxial in-plane tensile strains may be used to efficiently switch between the AFM multi-radical (i.e. spin polarised or open-shell) and diamagnetic quinoidal (i.e. closed-shell) states in a reversible manner. Such control arises from the effective manipulation of aryl ring dihedral angles, although other parameters such as the structural and chemical design of each 2D-CORF also play important roles. In addition, we demonstrate that the strain-induced electronic switching is robust at finite temperatures, which is a prerequisite for its experimental viability, and thus technological applicability. Overall, our work shows that integrating bi-radical moieties into covalent 2D materials is a promising route to achieve macroscopic control over molecular-scale electronic states.

## Results

**2D-CORF design and characterisation.** Following our previous work[25,30,34,35] and other studies[29] on these types of sytems, we employ the PBE0 hybrid density functional[38] (which incorporates

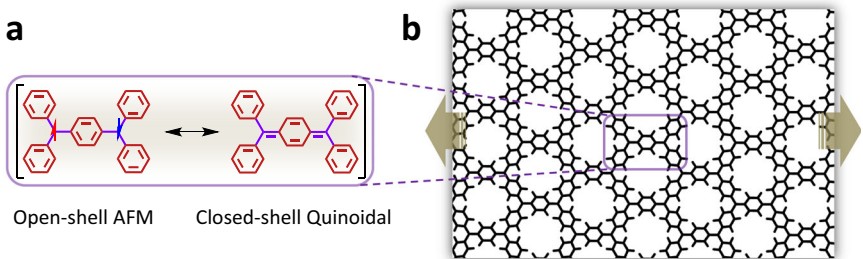

**Fig. 1 Bi-radical integration in 2D-CORFs. a** Schematic representation of the energetically close resonant valence bond forms in the ground state of Thiele's bi-radical: the open-shell AFM (left) and the closed-shell quinoidal (right). **b** 2D-CORFs viewed as extended 2D analogues of bi-radical compounds, where one could potentially switch between the two electronic states via external uniaxial strain (brown arrows).

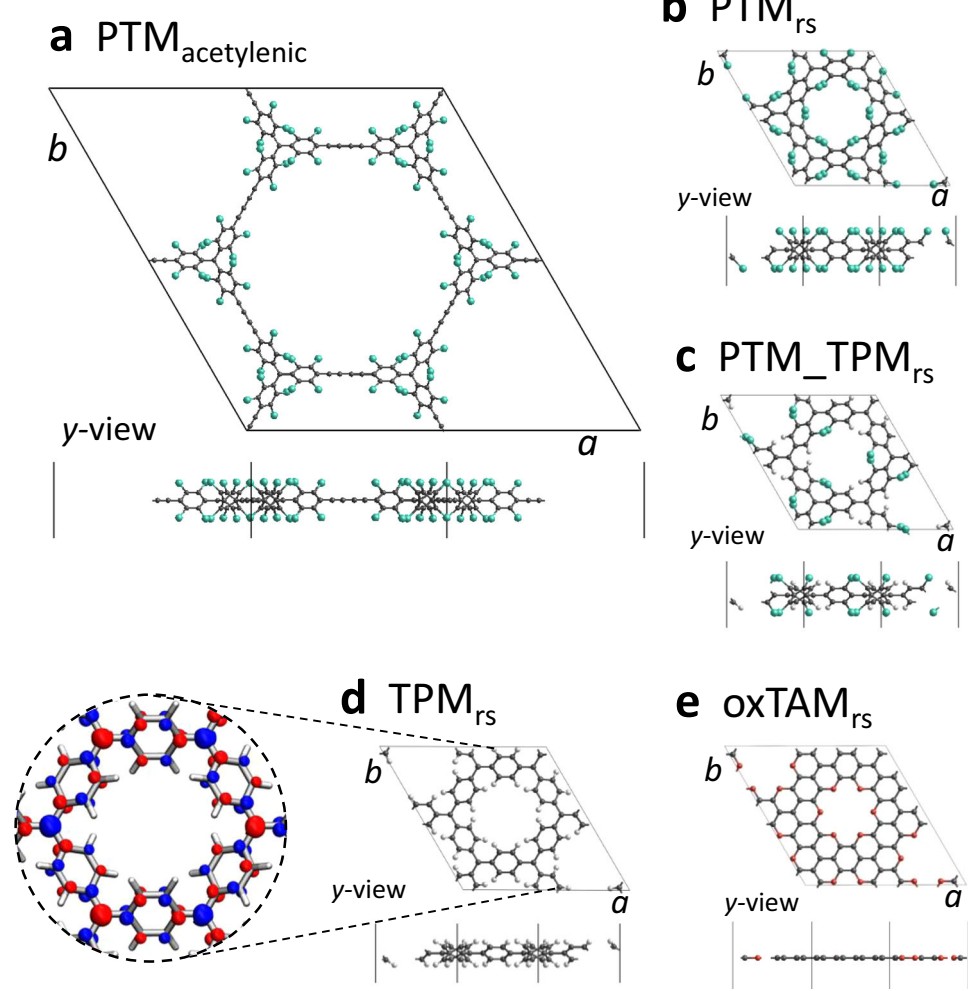

**Fig. 2 2D-CORF characterisation.** DFT-optimised crystal structures of **a** PTM$_{acetylenic}$, **b** PTM$_{rs}$, **c** PTM_TPM$_{rs}$, **d** TPM$_{rs}$ and **e** oxTAM$_{rs}$ 2D-CORFs. Colouring: C—dark grey, H—white, Cl—green, O—red. The spatially resolved spin density for the AFM solution (spin-up: blue, spin-down: red) is shown for TPM$_{rs}$ (**d**).

25% of non-local Hartree Fock exchange—HFE) in order to best capture the electronic structure of 2D-CORF materials. DFT-based calculations using hybrid functionals have proven to have remarkable reliability in describing the ground states and magnetic properties of organic radicals[39] and polyradicals[40–42]. The structures of all considered 2D-CORFs, which have been optimised using DFT/PBE0 calculations (see "Methods" for full details), are shown in Fig. 2. Our considered 2D-CORFs are hexagonal networks of sp$^2$ radical carbon centres (αC) covalently linked through π-conjugated groups[30]. The different linkers define the range of different 2D-CORFs that we consider: (i) the experimentally synthesised[32] PTM$_{acetylenic}$ (Fig. 2a, PTM stands for perchlorotriarylmethyl[43,44]), (ii) the chlorinated ring-sharing PTM$_{rs}$ (Fig. 2b), (iii) the ring-sharing mixed PTM_TPM$_{rs}$ (Fig. 2c, TPM stands for triphenylmethyl[45]), (iv) the ring-sharing TPM$_{rs}$ (Fig. 2d) and (v) the ring-sharing oxTAM$_{rs}$ (Fig. 2e, oxTAM stands for oxo-triarylmethyl[46]). PTM$_{acetylenic}$ (**2a**) has been experimentally characterised as an AFM semiconductor[32]. In PTM$_{rs}$ (**2b**), PTM_TPM$_{rs}$ (**2c**), TPM$_{rs}$ (**2d**) and oxTAM$_{rs}$ (**2e**), the αC centres are linked through a single aryl ring, thus significantly enhancing their electronic coupling as compared to PTM$_{acetylenic}$ (**2a**). Although such ring-sharing 2D-CORFs have not yet been experimentally reported, an analogue six-membered ring-sharing αC oligomer has been recently reported[47] and shown to possess a similar electronic structure to bi-radical compounds[18]. In the SI

(Section 3) we show that the experimentally derived nearest neighbour magnetic coupling between αC centres in this TPM$_{rs}$-like oligomer is very similar to that we predict for the TPM$_{rs}$ 2D-CORF, thus confirming the adequacy of our computational approach.

The different chemical functionalisation of the aryl rings in our considered 2D-CORFs determines their dihedral angles through steric hindrance[34]. Thus, through the series shown in Fig. 2 from **2a** to **2e**, we observe increasingly planarized aryl rings (see the mean dihedral angle in Table 1): from the highly twisted conformation in PTM$_{acetylenic}$ to the fully planar structure of oxTAM$_{rs}$, where oxygen atoms bonded between adjacent aryl rings fix them to be in-plane (see y-views in Fig. 2).

All 2D-CORFs exhibit a multi-radical open-shell ground state solution which is lower in energy than the closed-shell quinoidal minimum, except for TPM$_{rs}$, where both states are nearly degenerate (see $E_{AFM} - E_{QUI}$ in Table 1). The open-shell solution exhibits an alternating spin-up/spin-down ordering between αC centres, as shown with the spatially resolved spin density for TPM$_{rs}$ in Fig. 2d. Formally, this is a partial representation based on a broken symmetry (BS) solution resulting from the limitation of the single determinantal character of DFT-based calculations. Although the spin-polarisation ordering should not be taken directly at face value, such solutions are formally linked to open-shell states of the real system such as the FM or the AFM

**Table 1 Mean dihedral angles (degrees), the energy difference between AFM and closed-shell quinoidal (QUI) electronic solutions per $\alpha$C centre (meV), the average absolute value of spin populations on $\alpha$C centres in the AFM solution ($\langle|\mu_{\alpha C}|\rangle$) and the associated electronic bandgaps for all considered 2D-CORFs.**

|  | PTM$_{acetylenic}$ | PTM$_{rs}$ | PTM_TPM$_{rs}$ | TPM$_{rs}$ | oxTAM$_{rs}$ |
|---|---|---|---|---|---|
| Mean dihedral angle (degrees) | 48.0 | 46.8 | 40.1 | 32.4 | 0.0 |
| $E_{AFM} - E_{QUI}$ (meV) | −337.3 | −136.9 | −19.6 | 3.3 | −15.6 |
| $\langle|\mu_{\alpha C}|\rangle$ | 0.45 | 0.39 | 0.28 | 0.22 | 0.12 |
| Bandgap (eV) | 2.17 | 2.36 | 1.61 | 1.64 | 1.12 |

Note that, for oxTAM$_{rs}$, the $E_{AFM} - E_{QUI}$ value is taken as the difference between $E_{AFM}$ and the energy of the semimetallic solution (see SI Fig. S2).

states[48–51]. The degree of BS spin polarisation reflects the extent of the mixture between the open-shell and closed-shell valence bond forms. The appearance of such solutions also implies the emergence of low-lying states with non-zero net magnetisation and thus is a predictor of the existence of paramagnetism/antiferromagnetism for a particular system/condition. In order to quantify the magnitude of the AFM coupling in a periodic calculation one can assume that the magnetic system is described by a defined topology of localised spins and use the energy difference between the BS ground state solution and that of the FM state solution to map these states on to a Heisenberg model as described in the SI (see Table S2 and Section 3 in the SI)[52]. For simplicity, in the remainder, we will refer to BS open-shell solutions as AFM solutions, where we take the degree of spin polarisation in such solutions as an indicator of the strength of the open-shell character of the real state.

The increasing aryl ring twist angles and distance between radical centres, going from left to right through the series of 2D-CORFs in Table 1, also correlates with increasingly localised spin distributions (see values of the average of the absolute $\alpha$C spin population, $\langle|\mu_{\alpha C}|\rangle$). We also see that increased electronic localisation (e.g. due to more twisted aryl rings) results in a higher energetic cost associated with electron pairing (see $E_{AFM} - E_{QUI}$ in Table 1). For oxTAM$_{rs}$ we could not stabilise the localised quinoidal state which spontaneously falls into a fully delocalised semimetallic solution (see SI Fig. S2), in accordance with other fully planar 2D-CORFs (e.g. graphynes)[30]. The degree of electronic localisation in the AFM solution for each 2D-CORF is also reflected in the tendency for a decrease in the electronic bandgap (see Table 1) and an increase in band dispersion (see Fig. S1 in the SI) through the series of 2D-CORFs. These tendencies are also associated with the strength of the AFM coupling, which increases monotonically through the series of 2D-CORFs (see Table S2 in the SI).

As previously demonstrated for TPM$_{rs}$[30], the quinoidal closed-shell solution leads to a local pairing of electrons in specific aryl rings within the framework, which subsequently become more planar[30]. This structural response is due to the double bonds formed between the involved $\alpha$C centres and the aryl rings linking them. Herein, we explore the opposite phenomenon: i.e. whether by external manipulation of certain aryl rings we may induce electron pairing. To test this idea we consider the in-plane uniaxial strain of our 2D-CORFs[25,36]. Our starting electronic configuration is the unstrained open-shell AFM distribution shown in Fig. 2d with an associated alternating spin density. We highlight the response of TPM$_{rs}$ as a reference example of the prototypical structural response of our considered 2D-CORFs to uniaxial strain (see Fig. 3a). For the relaxed TPM$_{rs}$ conformation ($\varepsilon = 0\%$), all aryl rings are equally twisted ($\varphi_1 = \varphi_2 = 33°$). Upon stretching ($\varepsilon = 28\%$), the aryl rings parallel to the strain direction are flattened ($\varphi_2 = 4°$) and the remaining rings become twisted

out-of-plane ($\varphi_1 = 75°$). These conformational changes lead to a 12.5% reduction of the $b$ unit cell parameter (Fig. 3a).

These strain-induced conformational changes are likely to favour electron pairing within the flattened aryl rings ($\varphi_2$ in Fig. 3a). In principle, the higher the dihedral angle difference between the two types of aryl rings ($|\varphi_1 - \varphi_2|$) the higher the probability is to induce electron pairing. TPM$_{rs}$ displays the most significant conformational changes, followed by PTM_TPM$_{rs}$ and PTM$_{rs}$ (Fig. 3f), which is in accordance with the degree of chlorine functionalisation in the latter two materials, making their aryl rings more rigidly fixed. For PTM$_{acetylenic}$, the acetylenic linkers provide an extra degree of freedom which is not present in the other 2D-CORFs, leading to smaller changes of $|\varphi_1 - \varphi_2|$ with respect to strain. For the fully planar structure of oxTAM$_{rs}$, $|\varphi_1 - \varphi_2|$ remains invariable, and equal to zero, throughout the range of considered strain (Fig. 3f).

The tensile strength of each material, characterised by Young's modulus (YM), depends on the different structural degrees of freedom in each case. Figure 3g shows our calculated YM values for all 2D-CORFs in comparison with experimentally reported values for graphene[53] and single layer MoS$_2$[54]. Here, we can see that, except for the highly rigid oxTAM$_{rs}$, all 2D-CORFs have a YM that is both significantly lower than that of a typical inorganic 2D material and ~20 times smaller than that of graphene (~1000 Gpa[53]) helping to quantify the relative ease by which our 2D-CORFs can be strained. For the case of single-layer MoS$_2$, different on-substrate based methods have been employed to uniaxially stretch the material by ~2.5%[3]. Considering the magnitude of the in-plane YM of single-layer MoS$_2$ (see Fig. 3g), this implies that such experiments can readily apply in-plane uniaxial tensile stresses of at least 6.5 GPa. Taking TPM$_{rs}$ as an example, such tensile stress would induce a uniaxial strain of ~20%. In the case of more specialised experimental set-ups, graphene has been uniaxially stretched by almost 6%, showing that larger in-plane tensile stresses of up to ~60 GPa are also achievable[4]. We also note that the YM values of TPM$_{rs}$, PTM_TPM$_{rs}$, PTM$_{rs}$ and PTM$_{acetylenic}$ are all significantly smaller than that of oxTAM$_{rs}$ which strongly indicates that aryl ring twisting is the key factor leading to highly stretchable 2D-CORFs, rather than simply their nanoporous structure.

In order to test whether the strain-induced conformational changes depicted in Fig. 3 lead to a transition from the open-shell AFM electronic state towards the closed-shell quinoidal diamagnetic state, we have extracted $\langle|\mu_{\alpha C}|\rangle$ values throughout the full range of considered strains for the corresponding electronic solutions. Figure 4a shows the variation of $\langle|\mu_{\alpha C}|\rangle$ against uniaxial strain for each studied 2D-CORF. PTM$_{acetylenic}$ shows a high and robust $\langle|\mu_{\alpha C}|\rangle$ value throughout stretching, which is a consequence of the strongly localised unpaired electrons in this material, which in turn, is due to the large distance between $\alpha$C centres and the highly twisted aryl rings (48°). Conversely, oxTAM$_{rs}$ shows the

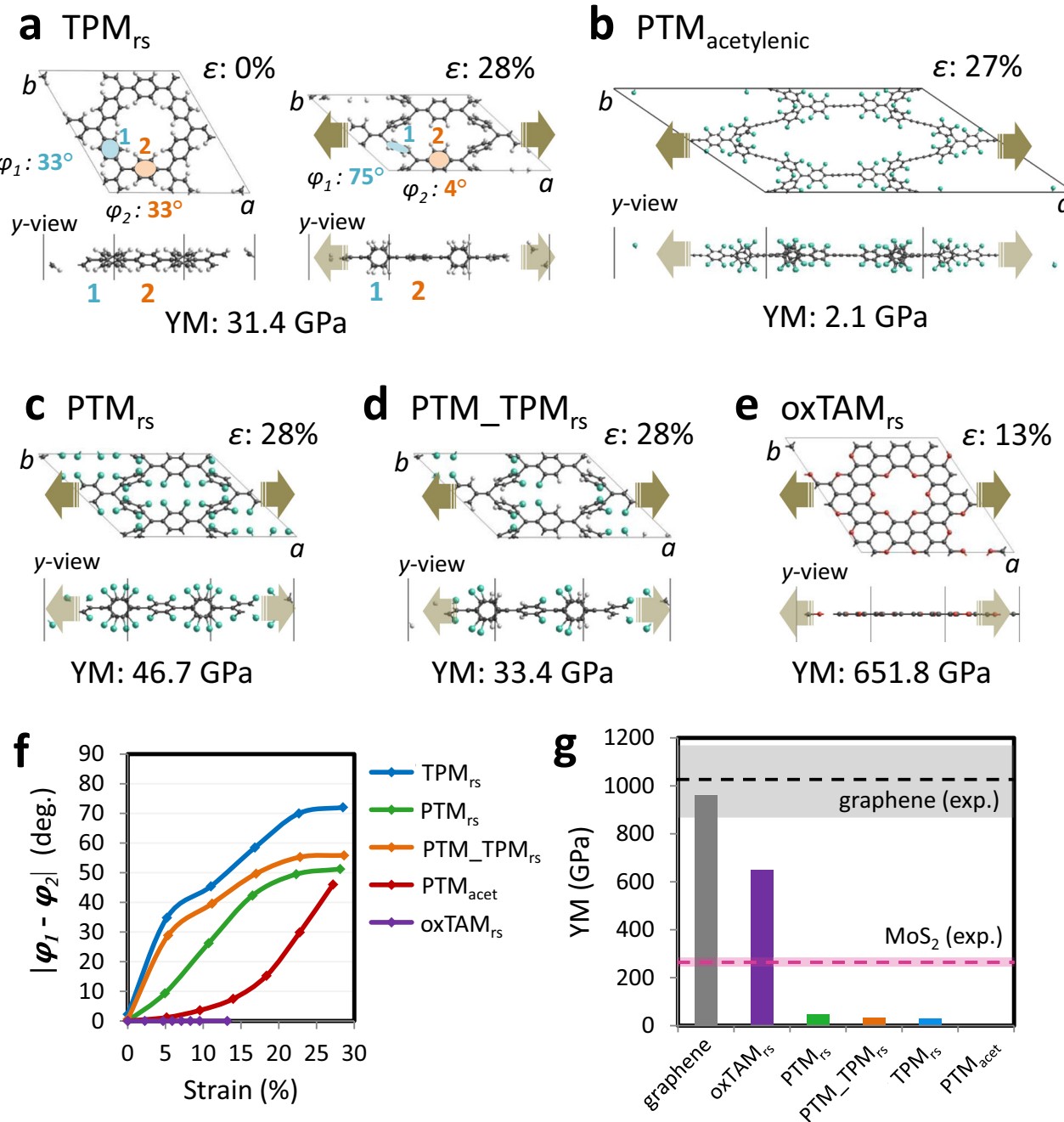

**Fig. 3 Structural response to strain. a–e** Crystal structures of the most stretched conformations for each of the studied 2D-CORFs indicating the associated strain ($\varepsilon$) and in-plane Young's modulus (YM). Atom colour key: C—dark grey, H—light grey, Cl—green, O—red. **f** Dihedral angle difference between the out-of-plane ($\varphi_1$) and in-plane ($\varphi_2$) twisted aryl rings versus uniaxial strain. **g** Calculated in-plane YM values for all 2D-CORFs and graphene (vertical bars) compared to experimental YM values of graphene[53] and single layer $MoS_2$[54] (horizontal dashed lines), where the associated range within which experimental values have been reported are indicated with coloured panels (see Table S1 in the SI for details).

lowest $\langle|\mu_{\alpha C}|\rangle$ value in the relaxed conformation. As previously mentioned, the aryl rings in $oxTAM_{rs}$ are fixed in-plane, and so uniaxial strain mainly induces stretching of π-π bonds along the strain direction. As a consequence, unpaired electrons in $oxTAM_{rs}$ become more localised, and thus $\langle|\mu_{\alpha C}|\rangle$ monotonically increases with increasing strain (Fig. 4a). In contrast, for $TPM_{rs}$, $PTM\_TPM_{rs}$ and $PTM_{rs}$, uniaxially straining the relaxed structure induces a clear transition from the open-shell AFM solution to the closed-shell diamagnetic quinoidal solution, in which $\langle|\mu_{\alpha C}|\rangle$ vanishes. $TPM_{rs}$, the 2D-CORF with the highest aryl ring twisting

flexibility, is most prone to electron pairing and exhibits a full depletion of $\langle|\mu_{\alpha C}|\rangle$ for strains between 5% to 22% (blue curve in Fig. 4a). For uniaxial strains above 25%, $\langle|\mu_{\alpha C}|\rangle$ rises again, which is a consequence of the associated αC-aryl ring bond distances reaching typical values of single carbon–carbon bonds (1.54 Å; see SI Fig. S3). In Fig. 4b–d, we show the three most representative situations, where one may clearly see the transition from the AFM open-shell solutions (see spin-density iso-surfaces in Fig. 4b, d) to the intermediate electron-paired quinoidal distribution (see highest occupied crystal orbital density in Fig. 4c). $PTM\_TPM_{rs}$ and $PTM_{rs}$

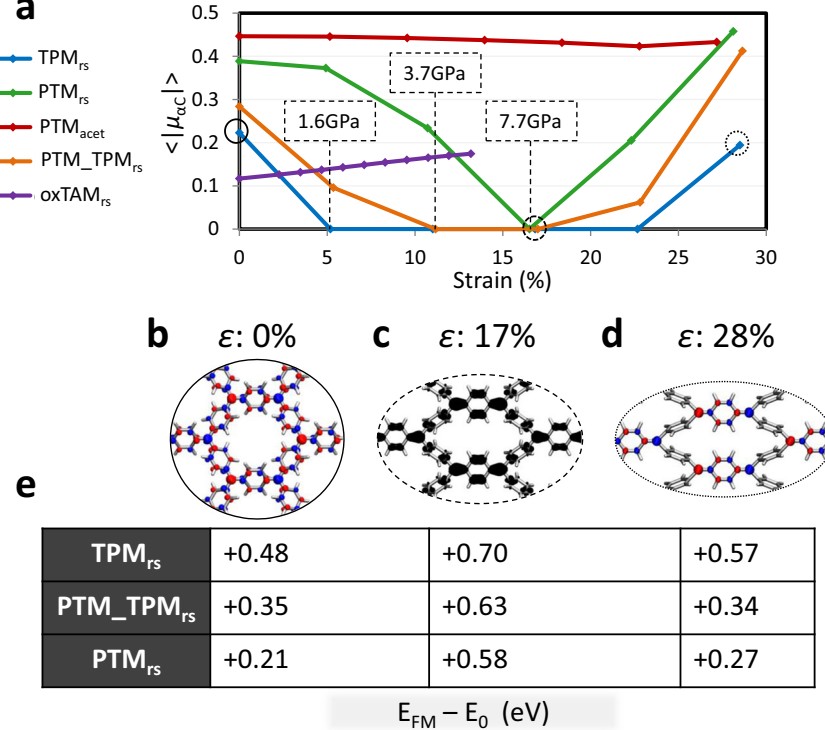

**Fig. 4 Electronic response to strain. a** Average of the absolute spin population per $\alpha C$ for each studied 2D-CORF versus uniaxial strain. Estimated in-plane tensile stresses required to induce the AFM-to-quinoidal transition are indicated for $TPM_{rs}$, $PTM\_TPM_{rs}$ and $PTM_{rs}$. **b** Spin density iso-surface (spin up: blue; spin down: red) for $TPM_{rs}$ when relaxed ($\varepsilon = 0\%$). **c** Highest occupied crystal orbital density (black) for the semi-strained conformation of $TPM_{rs}$ ($\varepsilon = 17\%$). **d** Spin density iso-surface for highly strained $TPM_{rs}$ ($\varepsilon = 28.5\%$). **e** Relative energy per $\alpha C$ (in eV) of the ferromagnetic electronic solution ($E_{FM}$) with respect to the electronic ground state ($E_O$) at each conformation (**b–d**) for $TPM_{rs}$, $PTM\_TPM_{rs}$ and $PTM_{rs}$. For the cases when the AFM solution is the ground state these energy differences are related to the degree of magnetic coupling (see also Table S2 in the SI).

also show a complete depletion of $\langle |\mu_{\alpha C}| \rangle$, which, in each case, is induced at a higher strain than for $TPM_{rs}$, and is maintained for a comparatively smaller range of strains (Fig. 4a). This behaviour is due to the more localised unpaired electrons in these two materials (see Table 1), and the lower aryl ring rotational flexibility arising from chlorine functionalisation (see Fig. 3f and SI Fig. S4). We note that the overall picture we obtain from the variation of $\langle |\mu_{\alpha C}| \rangle$ as a function of uniaxial strain (Fig. 4) is in agreement with that from bond length alternation (BLA) analysis (see SI Fig. S5), which is a structural indicator often used to characterise the balance between the AFM and quinoidal states in bi-radical compounds[22,55].

Although calculations on graphene have suggested that >15% in-plane biaxial strain could also induce AFM or quinoidal dimerised states[5], other theoretical studies point to the spontaneous rupture of the material at such strains[6]. Thus far, due to its ultrahigh in-plane strength, only tensile strains up to ~6% have been experimentally reached in graphene before failure[4]. In contrast, the relatively low tensile strengths of our 2D-CORFs permit the AFM to quinoidal interconversion via experimentally achievable relatively large elastic strains.

Finally, in Fig. 4e we provide the energetic cost associated with FM polarisation of the $\alpha C$ unpaired electrons (e.g. via external magnetic fields) with respect to the ground state at each structural conformation for the $TPM_{rs}$, $PTM\_TPM_{rs}$ and $PTM_{rs}$. This cost can be associated with the accessibility of open-shell states for a given system in a particular conformation. Here, we can see that the FM solution becomes significantly destabilised upon electron pairing, as induced in the three 2D-CORFs with uniaxial strains of ca. 17% (Fig. 4c). This extra energetic cost for closed-shell quinoidal diamagnetic ground states may be associated with the

process of breaking $\pi$–$\pi$ double bonds in order to generate a spin-polarised FM electronic distribution. This effect is particularly significant for $PTM_{rs}$ where the energetic cost to reach the FM solution is nearly three times larger for the "paired" configuration at $\varepsilon = 17\%$ as compared to the "unpaired" configuration at $\varepsilon = 0\%$. These results highlight the potential of ring-sharing 2D-CORFs as platforms with externally controllable spintronic characteristics.

Overall, the results of Fig. 4 demonstrate that at 0 K it is possible to induce electron pairing of $\pi$-conjugated electrons in 2D-CORFs by means of external uniaxial strain. Because of the conformational character of the mechanism (i.e. aryl ring twisting) and the effect of bond vibrations on electron delocalisation[34], it is expected that finite temperatures could affect the degree of external control over electron pairing. To test the robustness of our results at finite temperatures, we ran ab initio molecular dynamics simulations (AIMDS; see "Methods" for details) at 300 K for our three most promising 2D-CORFs, namely $TPM_{rs}$, $PTM\_TPM_{rs}$ and $PTM_{rs}$. As shown in SI Fig. S6, the strain-induced manipulation of aryl ring twisting previously characterised at 0 K (Fig. 3) holds at 300 K, despite fluctuations in dihedral angles induced by thermal vibrations. We note that there are some conformational differences between networks (see more details in Section 2 of the SI) but the overall structural response to strain is qualitatively the same for the three tested 2D-CORFs.

To assess whether the aryl ring twist manipulation also leads to a robust control over electron pairing, we have extracted the $\langle |\mu_{\alpha C}| \rangle$ values during the AIMDS at 300 K for the three 2D-CORFs at different strains. As one can see in Fig. 5, all 2D-CORFs show non-zero $\langle |\mu_{\alpha C}| \rangle$ values in the relaxed conformation ($\varepsilon = 0\%$; Fig. 5a, d, g). This is particularly interesting for $TPM_{rs}$ where the open-shell

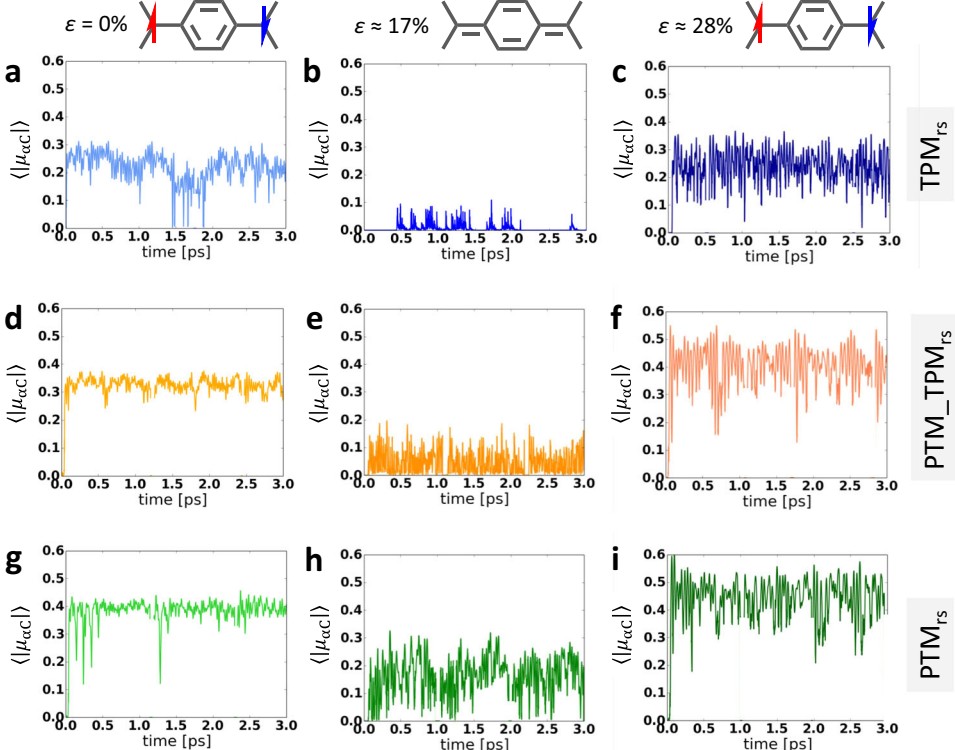

**Fig. 5 Electronic response to strain at 300 K.** Time-resolved evolution of the αC averaged absolute spin population $\langle|\mu_{\alpha C}|\rangle$ during 3 ps of AIMDS at 300 K for 0%, 17% and 28% uniaxial strains for (**a**–**c**) $TPM_{rs}$, (**d**–**f**) $PTM\_TPM_{rs}$ and (**g**–**i**) $PTM_{rs}$. The chemical sketches at the top of the figure represent the resulting dominant valence bond form for each strain.

AFM solution is slightly higher in energy at 0 K with respect to the closed-shell quinoidal one (see Table 1). It thus follows that finite temperatures, through bond vibrations inducing electron localisation[34], promote the open-shell AFM character (Fig. 5a). This is in good agreement with experimental variable-temperature measurements on the analogous six-membered $TPM_{rs}$ oligomer[47]. $\langle|\mu_{\alpha C}|\rangle$ is larger and more stable for $PTM\_TPM_{rs}$ and $PTM_{rs}$ in the relaxed conformations (Fig. 5d, g) due to the more perpendicular chlorinated aryl rings in these networks[34]. For $TPM_{rs}$, partial uniaxial strain ($\varepsilon = 16\%$) leads to full electron pairing, as demonstrated with the nearly complete depletion of $\langle|\mu_{\alpha C}|\rangle$, showing only a small degree of noise due to thermal fluctuations (Fig. 5b). Such effective pairing may be understood by the almost complete flattening of the aryl rings in $TPM_{rs}$ parallel to the strain direction (see SI Fig. S6b). $PTM\_TPM_{rs}$ and $PTM_{rs}$ (Fig. 5e, h) also display vanishing $\langle|\mu_{\alpha C}|\rangle$ values at $\varepsilon = 16\%$ (especially notable for $PTM\_TPM_{rs}$, Fig. 5e). However, in these cases, there is an increase of $\langle|\mu_{\alpha C}|\rangle$ noise due to thermal fluctuations, which is significantly detrimental in $PTM_{rs}$. Following the behaviour at 0 K, further stretching the networks again increases the $\langle|\mu_{\alpha C}|\rangle$ values for all materials (Fig. 5c, f, i), caused by the elongation of carbon–carbon bonds.

Overall, these results confirm the robustness of our proposed mechanical control of electron pairing at finite temperatures (300 K) via strain-induced manipulation of aryl ring twisting in 2D-CORFs. The chemical functionalisation of aryl rings is found to play a key role in determining aryl ring rotational flexibility under uniaxial strain (see Fig. S6 in the SI), in line with the response of insulating 2D-CORFs[36], and thus the effectiveness of electron pairing (Fig. 5). Thereby, $PTM\_TPM_{rs}$ appears to exhibit the most robust (and so potentially measurable) transition

between the open-shell AFM (Fig. 5d) and closed-shell quinoidal (Fig. 5e) electronic solutions. The behaviour of $PTM\_TPM_{rs}$ arises from a balance between electron localisation in the relaxed structure combined with significant conformational flexibility due to the mixed chlorine/hydrogen functionalisation of its aryl rings and is a promising candidate to be experimentally pursued.

## Discussion

In this work, we propose a strategy to externally control the transition between open-shell AFM and closed-shell quinoidal electronic states, which have long been studied in the field of molecular electronics[14–16]. Such states coexist in Kekulé organic bi-radicals[18] and have recently been reported in atomically precise graphene nanoribbons[56]. Although organic chemists have proven that chemical design is a powerful tool to induce one electronic state or another[16], a feasible procedure to achieve dynamic external control over electron pairing was lacking.

Here, we propose 2D-CORFs as ideal platforms to gain such control. We demonstrate that the application of uniaxial strain in 2D-CORFs allows one to effectively pair π-conjugated electrons within such otherwise open-shell multi-radical materials. The key factor for such strain-control comes through the mechanical manipulation of dihedral angles of aryl rings in these materials. Partial strain leads to a flattening of some aryl rings within the networks which, subsequently, leads to an effective electron pairing (quinoidalisation) within them. Further strain stretches the previously generated double bonds, which unpairs the electrons restoring the AFM spin distribution. Such an externally mediated electronic control mechanism is effective not only at 0 K but also at finite temperatures, as shown via AIMDS at 300 K. Our results also highlight the important role of structural and

chemical design of 2D-CORFs for enhancing stran-induced electronic control. In this respect, we demonstrate that radical centres need to be close to each other to achieve electron pairing and that the chemical functionalisation of aryl rings, determining the ease with which their dihedral angles may be manipulated, is an important factor.

Ongoing improvements in the bottom-up synthesis of single/ few-layer COFs with increasingly large crystalline domains will also assist the experimental realisation of our proposal[57]. The relatively low tensile strengths of our considered 2D-CORF materials also permit large elastic strains with experimentally accessible applied in-plane stresses, which is essential to ensure electronic control in real-world applications. Such high strains are not accessible in stiffer 2D materials, such as graphene, where similar strain-induced electronictransitions have been theoretically predicted[5] but which are experimentally intractable[4,6].

Overall, our proposal merges the fundamental switchability of bi-radicals[19–21] with the structural robustness and elasticity of organic 2D materials[10]. Unlike the effect of strain on most inorganic 2D materials, where minimal bond stretching induces small electronic changes[4–6], the remarkable efficiency of aryl ring twisting to induce electron pairing highlights the technological potential of rationally designed 2D organic materials.

## Methods

All 2D-CORFs studied in this work were fully optimised using periodic boundary conditions via an efficient cascade methodology. Classical force-field based geometry optimisations using the General Utility Lattice Programme[58] and the universal force field[59] were employed to generate pre-optimised structures (both atomic positions and cell-parameters). Then a second full geometry optimisation of both atomic structure and cell parameters was performed using DFT-based calculations and the PBE[60] functional with a Tier 1 light numerical atom-centred-orbital (NAO) basis set[61]. This was followed by a final full optimisation using the PBE0[16] hybrid functional (incorporating 25% HFE) with the same NAO basis set. In the SI we confirm that DFT calculations using the PBE0 functional are able to accurately to capture local magnetic coupling between $\alpha C$ centres in 2D-CORFs through comparison with results from experiments of a TPM$_{rs}$ oligomer. All DFT calculations employed a 6-6-1 gamma-centred Monkhorst Pack $k$-point sampling, except for the PTM$_{acetylenic}$ 2D-CORF which, due to its relatively large unit cell, was optimised using gamma–point sampling. Convergence criteria for these calculations were $1 \times 10^{-5}$ eV for total energy and $1 \times 10^{-2}$ eV/Å for the maximum residual force component per atom. We note that a Tier 1 light NAO basis set provides results of similar or higher quality to those obtained using a triple-zeta plus polarisation Gaussian type orbital basis set[62]. Details of how these NAO basis sets were developed are provided in ref. [63]. Single point PBE0 calculations were performed on these fully optimised structures, to generate band structures and isosurface maps of spin and crystal orbital electronic densities. Atomically partitioned spin densities on $\alpha C$ centres ($\mu_{\alpha C}$) were obtained via the Hirshfeld method[64]. The PBE0 geometry optimisations and single point calculations were performed separately for the open-shell AFM and closed-shell quinoidal solutions for each 2D-CORF. The open-shell AFM solutions were obtained with spin unrestricted DFT calculations and by setting an alternating spin polarisation initial guess over neighbouring $\alpha C$ sites. The closed-shell quinoidal solutions were obtained using closed-shell restricted DFT calculations. We provide an example of an input file (including the utilised basis set specification) used for our calculations in the SI, together with the cell parameters and atomic coordinates of fully optimised geometries for all studied 2D-CORFs in their relaxed conformation ($\varepsilon = 0$) for the open-shell AFM electronic solution.

To mimic the externally induced uniaxial strain on each 2D-CORF, we performed a series of restricted optimisations systematically increasing one of the cell parameters while allowing the other in-plane cell parameters and atomic positions to relax. Specifically, the $a(x)$ cell parameter was modified in steps of 0.2 Å for oxTAM$_{rs}$, 1 Å for PTM$_{rs}$, PTM_TPM$_{rs}$, and TPM$_{rs}$ and 2 Å for PTM$_{acetylenic}$.

The response of 2D-CORFs at finite temperatures was modelled using AIMDS. These were run for 3 ps at 300 K with fixed lattice parameters, using the Bussi–Donadio–Parrinello[65] thermostat, the hybrid PBE0 functional and a light (Tier-1) NAO basis set (see above). Further information on the post-analysis of AIMDS results is provided in Section 2 of the SI. All DFT-based calculations were performed using the FHI-AIMS code[63,66].

## Data availability

The data that support the findings of this study are available from the corresponding authors upon reasonable request. Optimised 2D-CORF geometries and FHI-AIMS input files used to generate much of the data used in this study can be found in the SI.

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

## Acknowledgements

This work was supported by MICIUN/FEDER RTI2018-095460-B-I00, PID2019-109518GB-I00 and CTQ2017-87773-P/AEI/FEDER (Spanish government, MINECO), MDM-2017-0767 ("María de Maeztu" programme for Spanish Structures of Excellence), 2017SGR13 and 2017SGR348 (Generalitat de Catalunya, DURSI). I.A. is grateful for support from the Alexander von Humboldt Foundation. R.S. acknowledges funding from MINECO under grant agreement FPI PRE2018-084053. The present work was performed using supercomputer resources as provided through grants from the Red Española de Supercomputación.

## Author contributions

I.A. and S.T.B. came up with the original concept and prepared the first version of the paper. I.A. carried out DFT (strain/AIMD) calculations on most of the studied 2D-CORFs and structural/electronic analysis for all materials. R.S. carried out DFT (strain) calculations and analysis of the mixed 2D-CORF and calculated the Young's modulus for all materials. J.R.A., M.D. and I.P.R.M. provided the magnetic analysis. I.P.R.M. performed the magnetic coupling calculations. All authors discussed the results and contributed to the paper preparation. S.T.B. coordinated the project.

## Competing interests

The authors declare no competing interests.
