## [Peer Review File · Nature Communications]

REVIEWER COMMENTS

Reviewer #1 (Remarks to the Author):

The authors claim to obtain switching between two states of two dimensional (2D) periodic conjugated organic networks, via quantum mechanical electronic structure computations. The two states are: a closed shell (non-magnetic) state and another with antiferromagnetic spin ordering. The basis of these two states rests with a valence bond (VB) representation of well known diradicaloid molecules where the two main VB states are a closed shell and a diradicaloid antiferromagnetic one. The novelty lies in the claim that this switching can be accomplished by anisotropic macroscopic strain applied externally to the 2D organic crystal networks.

The main concern, I fact, objection with this kind of computational modeling is the following. When the geometry of a molecule is stretched in the bond breaking direction, the electron correlation significantly changes, and a broken symmetry solution becomes more stable in a large number of one-electron theoretical descriptions. For instance, near the equilibrium geometry in the Hartree-Fock approximation, the solution is a closed shell one, while upon stretching, the HF solution becomes unstable for a spin polarized solution ("unrestricted Hartree-Fock"), which is lower in energy than HF. The latter breaks the symmetry. An example is that of the H₂ molecule. It should be clear to anyone, that when this symmetry breaking occurs, this only reflects the fact that the HF solution becomes unstable, due to the gradual change of the electron correlation, and not due to a real physical spin polarization.

The same is true in this case. Presumably, the "AFM" computations refer to spin-unrestricted DFT computations, often designated as UDFT, in analogy to spin-unrestricted Hartree-Fock computations. Stretching of the 2D network causes the one-electron single determinant description to become unstable with respect to spin polarization. This is not a real physical effect.

A further comment is that such an instability is highly dependent on the exact exchange component of the density functional applied. For this reason, even if the effect was real, one would obtain very different results depending on whether one uses different functionals, such as e.g. B3LYP, LDA, or any of the dozens of applicable functionals with exact exchange components ranging from $A_x=0$ to 1.

The uncertainty in any predicted Yang's modulus (Y) is increased by the approximation of the cross-section area, A in this article. This should normally be discussed. The resulting Y values are expected to be smaller than for graphene, because of the angle relaxations which are not available in graphene. The computed Y values appear reasonable.

Technical flaws include the lack of a detailed listing of key computational parameters, such that as required, the computations could be in principle be repeated. The choice of the Brillouin zone integration, the basis set (the reader is given a hint only: "a light numerical basis set"), the criteria for the geometry optimization convergence. These are normally included in the Supporting Information section.

The force field computations are largely separate from the rest of the paper. These are not very well justified given the delocalized nature of the electronic structure. The force field is not defined at all. The authors mention "... an efficient cascade methodology" without any referencing. Such information is normally provided in detail as part of Supporting Information.

A few missing references from the vast literature dealing with spin polarization of organic low

dimensional materials.

Mataga, N., *Theor. Chim. Acta* 1968, 10, 372.

Ovchinnikov, A. *Theor. Chim. Acta*. 1978, 47, 297-304.

Rajca, A., 1999. *High-Spin Polyradicals* (p. 345). Marcel Dekker: New York.

Reviewer #2 (Remarks to the Author):

Overall, the paper is very high-quality and appropriate for the journal.

It is ready to be published after minor corrections.

The Paper is stimulating and original.

Results are reliable.

The authors say that they want to use strain manipulation in future electronic devices, perhaps they could expand on a scheme for this or existing mechanisms for this.

The basis set for the DFT calculations should be discussed and specified in more detail, including for the molecular dynamics. More computational details could be included in the SI.

Reviewer #3 (Remarks to the Author):

The authors present a careful and compelling computational study of a series of Covalent Organic Framework (COF) materials whose linkers are designed to incorporate bi-radicals. A feature of these bi-radicals is that they give rise to two energetically close states: one open-shell antiferromagnetic and the other closed-shell diamagnetic. The key contribution of this work is that the authors suggest a promising means to efficiently switch between these two states at a scale that would be useful at the device level. It is correctly emphasised that spin-state switching in molecules and materials has been discussed for a while, but that this discussion usually lacks a viable suggestion of how these phenomena could be leveraged at scale.

Specifically, they demonstrate that applying in-plane strain along a particular axis of the material causes structural deformations that promote the population of the closed-shell diamagnetic state relative to the antiferromagnetic state. Further, they also show that the strain required to induce this electronic behaviour is more likely to be possible for these materials than, for example, graphene.

I believe this work is thought-provoking and of broad interest, and so I recommend the publication of this paper, provided that the referees can respond to the following:

- how confident are the authors in their chosen electronic structure method? I ask because, while the predicted ground state for the PTMacetylenic material matches experiment, the predicted energy difference between the AFM and quinoidal states for this material is significantly larger than for the other materials (particularly the two most promising ones in this study, for instance the

energy difference for PTM_TPM-rs is less than 0.02 eV). If the chosen hybrid DFT method systematically leads to the false ground state in these cases, then the switching behaviour could not be observed through the mechanism for the materials described in this paper. Is there some reference method that could be applied in a single point calculation to some of these materials to solidify this section?

That said, I do feel that, in any subsequent attempt to realise such a material, this is simply a matter of material design and that the mechanism to achieve state switching described by the authors is still of value.

Finally, some minor corrections and questions:

- It is a shame that the state-switching effect is not predicted for the PTMacet material, which has already been synthesised. Given the potential interest here, and that there are some design principles that have been identified, have the authors thought about how their computational workflow could be used to identify promising materials through some targeted screening? It would be exciting to see such a computationally led effort.
- Authors should define what they mean by “top down” and “bottom up” in the introduction.
- An explanation of what the arrows mean in fig 1b should be added to the caption so that they may be understood without reference to the main text.
- The shading in fig 3g is an imprecise way of showing the error of these measurements on this graph. Could these shaded areas be replaced with some low-opacity panes for example?
- Some of the y-view images in figure 2 are difficult to see. It would be good if these could be made clearer.
- I would encourage the authors to provide more detail on their computational methodology. Specifically:
 1. Structures in a machine-readable format.
 2. Input files for electronic structure codes to reproduce the values we see in the paper from the above structures.

Response to reviewers:

Reviewer 1:

Reviewer 1 makes some general comments on the modelling approach we employ. Most of the technical points raised are quite valid and are well taken. It is true that results of reported DFT calculations are often taken at face value with little thought as to their origin. We would thus like to thank the reviewer for nudging us into providing a fuller justification of our calculations. However, this said, in the present case the points raised either: i) do not apply directly to the systems we model, and/or ii) do not mean that we cannot extract a valid physical interpretation of our results. We clarify these issues in our response below and in the revised manuscript.

“When the geometry of a molecule is stretched in the bond breaking direction, the electron correlation significantly changes, and a broken symmetry solution becomes more stable in a large number of one-electron theoretical descriptions. For instance, near the equilibrium geometry in the Hartree-Fock approximation, the solution is a closed shell one, while upon stretching, the HF solution becomes unstable for a spin polarized solution (“unrestricted Hartree-Fock”), which is lower in energy than HF.”

Agreed. The single determinant closed-shell description is valid in the region near the equilibrium geometry since it dominates the wave-function of the ground state. Other valence-bond (VB) forms (biradical for homolytic dissociation, charge-transfer or ionic for heterolytic dissociation) start to compete and even dominate at longer distances.

“The latter breaks the symmetry. An example is that of the H₂ molecule. It should be clear to anyone, that when this symmetry breaking occurs, this only reflects the fact that the HF solution becomes unstable, due to the gradual change of the electron correlation, and not due to a real physical spin polarization.”

We agree that the spin polarisation in such broken symmetry (BS) solutions should not be taken directly as a physical spin polarisation. However, this does not mean that broken symmetry solutions are without physical meaning. At large internuclear separations, H₂ behaves like a diradical with characteristic low-lying singlet-triplet states. Here, it has been long established that the BS solution corresponds to a mixture of singlet and triplet states with an energy half-way between these two (see: J. Chem. Phys. 1981, 74, 5737; Chem. Phys. 1986, 109, 131). At intermediate separations, the ground state is a combination of open-shell and closed-shell VB forms. Here, BS solutions correspond to variational spin-polarised single determinant solutions to the Hartree-Fock equations and are related to

a mixing of different states. The BS spin polarisation reflects the extent of the mixture between the closed-shell VB form and the open-shell forms (i.e. singlet and triplet). It has been established that a formal mapping exists between the energy of a spin polarised BS solution and that of the open-shell singlet state of the real system (see: J. Phys. Chem. A 1997, 101, 705; J. Phys. Chem. A, 1997, 101, 7860). For a BS solution, the spin polarisation and the energetic proximity of a triplet state are clear indications of the open-shell nature of the ground state. As such, BS solutions are standardly used to describe open-shell molecular states (see: J. Phys. Chem. A 1997 101, 7860).

For periodic systems, a similar approach can be used to represent key spin distributions (the FM and simple AFM solutions) to estimate the relevant magnetic exchange interactions between localised magnetic moments (see: Phys. Chem. Chem. Phys., 2006, 8, 1645). For our 2D-CORFs, an open-shell BS ground state solution thus implies the presence of energetically low-lying solutions with non-zero net magnetization. The extreme of these is the pure FM solution, which can be well described by single determinant approaches. The energy of the FM solution (see Fig. 4e and Table S2), together with that of the BS solution, can be mapped onto energies of actual magnetic states to estimate the magnetic coupling (J) predicted by the DFT calculations (see also below).

Accordingly, an open-shell BS ground state solution strongly suggests that the material will display a paramagnetic response (or a weak paramagnetic/AFM response if $|J|$ is of the order of $k_B T$). This is in stark contrast with a closed-shell solution, where the excited states with non-zero net magnetization lay relatively high in energy and so they are not accessible (e.g. via room temperature or moderate external magnetic fields). Therefore, the transition from an open-shell solution to a closed-shell solution observed in our calculations upon stretching is indicative of a significant change in observable magnetic properties. In the manuscript we separately discuss the lowest-lying electronic open-shell solutions and the associated observable consequences according to this representation of the magnetic states of the systems.

“The same is true in this case. Presumably, the “AFM” computations refer to spin-unrestricted DFT computations, often designated as UDFT, in analogy to spin-unrestricted Hartree-Fock computations. Stretching of the 2D network causes the one-electron single determinant description to become unstable with respect to spin polarization. This is not a real physical effect.”

Indeed, we use the standard terminology of BS solutions to designate the UDFT description of the lowest energy open-shell electronic solutions. However, in our systems we observe the opposite effect to that described by the reviewer. Our calculations predict that our CORFs exhibit an open-shell solution when unstrained. This is in line with the experimental reports for the $\text{PTM}_{\text{acetylenic}}$ CORF, which is observed to have an AFM ground state (See refs. 31 and 32). Upon straining the CORFs, a closed-shell diamagnetic solution (well described by a single determinant) is predicted to be energetically more stable. This effect is due to a sterically induced twisting of the aryl rings in these materials, which enhances pairing of electrons between alpha-C centres. This novel effect is realised at relatively low strains. Only, at considerably larger strains do we observe strain-induced re-emergence of an open-shell BS solution for some systems. As discussed above, this BS solution indicates that paramagnetism is favoured at high strain.

“A further comment is that such an instability is highly dependent on the exact exchange component of the density functional applied. For this reason, even if the effect was real, one would obtain very different results depending on whether one uses different functionals, such as e.g. B3LYP, LDA, or any of the dozens of applicable functionals with exact exchange components ranging from $A_x = 0$ to 1.”

We agree with reviewer 1 that the choice of functional is very important in describing our systems accurately. Therefore, the use of the PBE0 functional, which contains 25% HF-like exchange, has been validated against experimental data prior to be applied to study our 2D-CORFs. As shown in our previous work (ref. 35) and in other studies (ref. 29), PBE0 calculations provide a ground state solution that is in line with experiment for the $\text{PTM}_{\text{acetylenic}}$ CORF. To provide further confirmation of the suitability of this choice of functional, we compare our calculated J values for the TPM_{rs} CORF with a J

value derived from experimental measurements on a superbenzene molecule (a ring fragment of the extended TPM_{rs} CORF). This detailed comparison is included in the revised SI. The obtained good agreement further confirms the suitability of the PBE0 functional for our systems.

“Technical flaws include the lack of a detailed listing of key computational parameters, such that as required, the computations could be in principle be repeated. The choice of the Brillouin zone integration, the basis set (the reader is given a hint only: “a light numerical basis set”), the criteria for the geometry optimization convergence. These are normally included in the Supporting Information section.”

We now provide all necessary computational parameters in order to repeat our DFT calculations. In addition, we provide an example of the FHI-AIMS input file we use, including a full specification of the numerical basis set, in the SI.

“The force field computations are largely separate from the rest of the paper. These are not very well justified given the delocalized nature of the electronic structure. The force field is not defined at all. The authors mention “... an efficient cascade methodology” without any referencing. Such information is normally provided in detail as part of Supporting Information.”

Details of the force field (Universal Force Field - UFF) are now provided. The UFF calculations are simply used to give reasonable pre-optimised CORF structures as inputs for subsequent DFT-based optimisation to make the calculations computationally more efficient. We have previously used UFF calculations to study the structure and energetics of other strained π -conjugated 2D-CORFs – where we show good agreement with DFT calculations (see ref. 36).

“A few missing references from the vast literature dealing with spin polarization of organic low dimensional materials. Mataga, N., Theor. Chim. Acta 1968, 10, 372. Ovchinnikov, A. Theor. Chim. Acta. 1978, 47, 297-304. Rajca, A., 1999. High-Spin Polyradicals (p. 345). Marcel Dekker: New York.”

The three suggested references have been included.

Reviewer 2:

“Overall, the paper is very high-quality and appropriate for the journal. It is ready to be published after minor corrections. The Paper is stimulating and original. Results are reliable.”

We thank the reviewer for his/her very positive evaluation of our work.

“The authors say that they want to use strain manipulation in future electronic devices, perhaps they could expand on a scheme for this or existing mechanisms for this.”

In addition to the cited review on experimental strain engineering of 2D materials (i.e. ref. 3), we now include a citation to a more recent review (D. Zhaohe et al. Strain Engineering of 2D Materials: Issues and Opportunities at the Interface. Adv. Mater. 2019, 31, 1805417). Following this work, we now include new text in the manuscript to highlight that strain in 2D materials can be achieved in a number of ways: i) growing the 2D material on epitaxial substrates with a controlled lattice constant mismatch; ii) thermal-expansion mismatch between the 2D material and its substrate; and iii) transferring the 2D material on a flexible substrate and directly stretching, compressing or bending the substrate.

“The basis set for the DFT calculations should be discussed and specified in more detail, including for the molecular dynamics. More computational details could be included in the SI.”

We now provide all computational details required for repeating our calculations in the methods section. We also include an example of the FHI-AIMS input file we use (with the full basis set) and atomic coordinates of all unstrained CORFs in the revised SI.

Reviewer 3:

"I believe this work is thought-provoking and of broad interest, and so I recommend the publication of this paper, provided that the referees can respond to the following:

- how confident are the authors in their chosen electronic structure method? I ask because, while the predicted ground state for the PTMacetylenic material matches experiment, the predicted energy difference between the AFM and quinoidal states for this material is significantly larger than for the other materials (particularly the two most promising ones in this study, for instance the energy difference for PTM_TPM-rs is less than 0.02 eV). If the chosen hybrid DFT method systematically leads to the false ground state in these cases, then the switching behaviour could not be observed through the mechanism for the materials described in this paper. Is there some reference method that could be applied in a single point calculation to some of these materials to solidify this section?"

We thank the reviewer for his/her positive evaluation of our work and suggestions that have improved its overall readability and presentation. Regarding the above point, to allow for efficient strain-induced conformational switching between distinct electronic states, it is likely that the two states should lay close in energy. Indeed, as Reviewer 3 mentions, both PTM_TPM_{rs} and TPM_{rs} CORFs are examples where we find a small energy difference between open-shell and quinoidal solutions and efficient switching. In order to confirm that our predictions in such cases are as robust as possible, we compared our calculated values of the AFM coupling (J) for the TPM_{rs} CORF with a J value derived from experimental measurements on a superbenzene molecule (a ring fragment oligomer of the extended TPM_{rs} CORF). The obtained good agreement provides further confirmation of the suitability of our electronic structure method (i.e. DFT with the PBE0 hybrid functional) for these systems. We also note that in the case of PTM_{rs} the unstrained open-shell versus quinoidal energy difference is significantly larger than for PTM_TPM_{rs}, but conformational switching between the two solutions is still observed. This detailed comparison is now included in the revised SI.

"That said, I do feel that, in any subsequent attempt to realise such a material, this is simply a matter of material design and that the mechanism to achieve state switching described by the authors is still of value."

We agree, and indeed it was our intention, that the main message of our work is to confirm the viability of a new materials-level mechanism for conformationally biasing either open-shell or closed-shell states

"Finally, some minor corrections and questions:

- It is a shame that the state-switching effect is not predicted for the PTMacet material, which has already been synthesised. Given the potential interest here, and that there are some design principles that have been identified, have the authors thought about how their computational workflow could be used to identify promising materials through some targeted screening? It would be exciting to see such a computationally led effort."

We are now working on virtual screening for CORFs that display particularly efficient and robust strain-induced switching for device applications. For this we plan to follow-up our previous study (ref. 36) where we use computationally efficient force field calculations to quickly establish the strain-induced response of many CORFs (followed by DFT calculations on promising materials). We are focusing on CORFs which have suitable internal steric interactions, and which maintain planarity when strained.

"- Authors should define what they mean by "top down" and "bottom up" in the introduction."

Top-down 2D materials: via exfoliation of single layers from a bulk layered material. Bottom-up 2D materials: via synthesis from molecular building blocks. This is now clarified in the text.

"- An explanation of what the arrows mean in fig 1b should be added to the caption so that they may be understood without reference to the main text."

Done.

"- The shading in fig 3g is an imprecise way of showing the error of these measurements on this graph. Could these shaded areas be replaced with some low-opacity panes for example?"

We note that the shading was not used to show measurement errors, but an approximate range of values reported in the literature. In any case we have now used panes to indicate this uncertainty, and further clarified their meaning in the figure caption.

"- Some of the y-view images in figure 2 are difficult to see. It would be good if these could be made clearer."

Done.

"- I would encourage the authors to provide more detail on their computational methodology. Specifically:

1. Structures in a machine-readable format.

2. Input files for electronic structure codes to reproduce the values we see in the paper from the above structures."

All computational details required to reproduce our results are now provided in the methods section. We also provide an of the example FHI-AIMS input file we use (with the full basis set) and atomic coordinates of all unstrained CORFs in the revised SI document.

REVIEWERS' COMMENTS

Reviewer #1 (Remarks to the Author):

I am skeptical with respect to the main tenets of this article. These molecular materials are typically non-crystalline. It may be possible perhaps at some point to make single layers of a significant size of these materials on the top of a crystal surface. Then, however, how do you place these layers under a uniaxial strain of such an enormous magnitude that is apparently required to switch their ground states?

This is highly speculative, and therefore I am not recommending publication in Nat. Comm.

Reviewer #2 (Remarks to the Author):

The authors have responded to and incorporated the suggestions of the referees.

The paper is now ready for publication.

Reviewer #3 (Remarks to the Author):

The authors have addressed all of the questions and concerns that I initially raised.